# Improved UNet with Attention for Medical Image Segmentation

**DOI:** 10.3390/s23208589

**Published:** 2023-10-20

**Authors:** Ahmed AL Qurri, Mohamed Almekkawy

**Affiliations:** School of Electrical Engineering and Computer Science, Pennsylvania State University, University Park, PA 16802, USA; mka9@psu.edu

**Keywords:** UNet, UNet++, Transformer, CNN, attention, medical imaging, ultrasound, CT scan

## Abstract

Medical image segmentation is crucial for medical image processing and the development of computer-aided diagnostics. In recent years, deep Convolutional Neural Networks (CNNs) have been widely adopted for medical image segmentation and have achieved significant success. UNet, which is based on CNNs, is the mainstream method used for medical image segmentation. However, its performance suffers owing to its inability to capture long-range dependencies. Transformers were initially designed for Natural Language Processing (NLP), and sequence-to-sequence applications have demonstrated the ability to capture long-range dependencies. However, their abilities to acquire local information are limited. Hybrid architectures of CNNs and Transformer, such as TransUNet, have been proposed to benefit from Transformer’s long-range dependencies and CNNs’ low-level details. Nevertheless, automatic medical image segmentation remains a challenging task due to factors such as blurred boundaries, the low-contrast tissue environment, and in the context of ultrasound, issues like speckle noise and attenuation. In this paper, we propose a new model that combines the strengths of both CNNs and Transformer, with network architectural improvements designed to enrich the feature representation captured by the skip connections and the decoder. To this end, we devised a new attention module called Three-Level Attention (TLA). This module is composed of an Attention Gate (AG), channel attention, and spatial normalization mechanism. The AG preserves structural information, whereas channel attention helps to model the interdependencies between channels. Spatial normalization employs the spatial coefficient of the Transformer to improve spatial attention akin to TransNorm. To further improve the skip connection and reduce the semantic gap, skip connections between the encoder and decoder were redesigned in a manner similar to that of the UNet++ dense connection. Moreover, deep supervision using a side-output channel was introduced, analogous to BASNet, which was originally used for saliency predictions. Two datasets from different modalities, a CT scan dataset and an ultrasound dataset, were used to evaluate the proposed UNet architecture. The experimental results showed that our model consistently improved the prediction performance of the UNet across different datasets.

## 1. Introduction

Medical image segmentation plays a crucial role in the detection of various diseases and tumors. Two of the most-widely used medical imaging techniques are ultrasound and Computed Tomography (CT) [1]. Traditionally, the delineation of pathological structures from medical images has been performed manually. This procedure is exceedingly time-consuming and requires suitable clinical expertise to obtain clinically relevant contours. The increase in medical imaging data that require interpretation by clinicians for diagnosis or surgical purposes has paved the way for the development of computer-aided image analysis tools [2]. Mathematical models that incorporate statistical shapes and low-level image processing have been widely used for segmentation [3]. Other examples of mathematical approaches include cluster-based algorithms such as K-means [4] and an active contour model based on the local and global intensities [5]. However, issues such as differences in tissue appearance, low resolution, and weak boundaries make these methodologies less robust to noise and non-uniform contrasts in medical imaging [3,6]. More recently, Deep Learning (DL) techniques have been successful in a wide range of computer vision applications such as object recognition and semantic segmentation [1]. Additionally, these techniques, particularly semantic segmentation, have been successfully applied in the medical image domain. Existing medical image-segmentation methods primarily employ Convolutional Neural Networks (CNNs) [7] partly due to their fast inference speed and powerful feature representation [8]. Convolutional operations extract local features from an image by extracting local characteristics from neighboring pixels [9]. A variety of segmentation models have been developed based on CNNs such as Fully Convolutional Networks (FCNs) [10], UNet [11], UNet 3+ [12], and DeepLab [13], among others [14,15,16,17,18,19,20,21,22,23,24]. UNet is one of the earliest and most-widely used techniques in medical image segmentation developed by Ronneberger et al. [11] based on an encoder–decoder architecture [12]. It is composed of a contracting path for downsampling and an expansive path for upsampling with skip connections in between to enhance the detail lost during downsampling. Ronneberger et al. originally employed the UNet architecture for Electron Microscopy Image (EM) segmentation, as part of the International Symposium on Biomedical Imaging (ISBI) 2012 challenge [11]. Many applications of UNet followed this; for example, Leclerc et al. used UNet to segment a cardiac ultrasound dataset; the work of [25,26] used a MultiRes UNet for carotid–artery–stenosis diagnosis [27] for the automatic extraction of muscle parameters. However, UNet has several shortcomings. For instance, the direct skip connections in UNet result in a large semantic difference between the input of the two convolutional layers [28]. Moreover, skip connections join feature maps from the same scale without considering the relationship between feature maps from different stages. This results in feature representations that may not be consistent [9]. To remedy these problems, numerous enhancements and modifications have been made to UNet. One of the most-notable developments is UNet++, which introduces nested and dense skip connections between different stages using different shortcut connections to reduce the semantic gap between the encoder and decoder [29].This results in a reduction in the semantic gap between the encoder and decoder to capture deeper contextual representations [9]. Since UNet++ is also based on CNNs, it suffers from limitations associated with CNNs, as will be elaborated upon in the following sections.

Deep supervision was introduced to improve performance by enabling intermediate layers to learn discriminative features and solve the vanishing gradient issue [30]. BASNet is an example of a network architecture that employs deep supervision via side-outputs and was initially proposed for salient object detection [31]. Moreover, the introduction of an attention mechanism is another improvement that assists in overcoming the constraints of convolutions [32]. The attention mechanism has been proposed to mimic the human visual system by concentrating on a portion of the most-relevant information. These attention techniques can be categorized into four groups. Channel attention resembles what to focus on, while spatial attention resembles the location of the focus. Branch attention, on the other hand, resembles what to focus on, and hybrid attention includes a combination of channel and spatial attention [2]. An example of channel attention is Squeeze-and-Excitation (SE) [33]. The SE block exploits inter-channel dependencies using a squeeze operation followed by an excitation function. Furthermore, the introduction of Attention Gates (AGs) by Oktay et al., originally used to segment the pancreas [34], is another form of attention. The AG focuses on crucial features pertinent to a certain task and hides less-relevant features [35]. Similarly, Channel-UNet [36] introduced spatial channelwise convolution to recalibrate the spatial and channel-level features. Additionally, the Convolutional Block Attention Module (CBAM) [37] is an attention module that applies attention to both spatial and channel dimensions. Lastly, SCAUNet [38] employs both spatial and channel attention and integrates them as a plug-and-play module.

Although the CNN-based technique is highly successful, it has limitations in modeling long-range dependencies owing to the inherent inductive biases [39]. Furthermore, the pooling and convolution layers might prevent low-level features from being propagated to the next convolutional layers [9]. The Transformer offers the advantage of modeling global information using self-attention mechanisms, a technique that was originally employed for sequence-to-sequence tasks such as Natural Language Processing (NLP) [40]. One of the notable works is the Vision Transformer (ViT) model [41], which employs the Self-Attention mechanism (SA), which has attained state-of-the-art results in ImageNet classification [42]. ViT simultaneously utilizes multiple self-attention heads to capture long-range dependencies [43]. However, Transformers also lack detailed localization information [44]. To capture low-level details and long-term dependencies simultaneously, networks such as TransUNet [45] and TransAttUNet [9] have proposed the use of a hybrid CNN-Transformer for medical image segmentation. It employs CNNs to extract low-level spatial information and a Transformer to learn global information, followed by upsampling [45]. Conversely, Swin-UNet removes the CNN and employs a complete Transformer structure using a shifted window mechanism to extract low-level details and a patch-expanding layer for upsampling [46]. Wang et al. [8] argued that TransUNet, similar to the UNet architectures, has skip connections at the same level, which constrains feature fusion. Instead, new model architectures such as MS-TransUNet++ [8] and CoT-UNet++ have been proposed [47]. These models use dense skip connections between the encoder and decoder at different levels to improve feature fusion, similar to UNet++, in addition to a hybrid encoder that includes a Transformer. For example, MS-TransUNet++ utilizes a Transformer to extract global information, similar to TransUNet, in addition to dense skip connections. Moreover, MS-TransUNet++ employs a weighted loss function that combines focal loss, Jaccard index loss, and multiscale structure similarity loss [8].

Another noteworthy research relevant to this study is the TransNorm network. The decoder and skip connections were enhanced by adding a spatial normalization module from Transformer [42]. Azad et al. achieved great accuracy using TransNorm in segmenting medical imaging datasets such as the International Skin Imaging Collaboration (ISIC) and the Multiple Myeloma (MM) segmentation challenge [42].

In this paper, we propose an improved UNet architecture for the segmentation of medical imaging. As mentioned earlier, the nature of medical imaging makes segmentation challenging due to poor image quality, blurry boundaries, and textures [48]. Ultrasound images, in particular, suffer from speckle noise, shadows, and signal dropout. Our model helps overcome these issues by considering the complementary properties of CNNs and Transformer, but we also focused on improving the feature representation from skip connections to benefits from this hybrid architecture. First, to extract the long-range contextual representation, a Transformer was added at the bottleneck as a bridge between the encoder and decoder. This hybrid CNN-Transformer network architecture utilizes the spatial feature map from the CNN and the global context from the Transformer [45]. Most importantly, on the decoder side, a Three-Level Attention (TLA) module was devised in the decoder stages similar to the two-level Attention Gate in TransNorm [42], but with an additional Attention Gate (AG). Zhang et al. [49] used AG at the end of the skip connections before the decoding stage to preserve structural information and to enhance information fusion. The AG mechanism is also helpful for the model to focus better on salient features. Furthermore, the TLA module uses the Transformer spatial normalization output to enrich the feature map originating from the skip connections. Additionally, the attention mechanism allows the network to adjust the weights dynamically based on the most-relevant features [50].

Instead of using simple, direct skip connections with TLA, we opted for dense skip connections that are similar to UNet++ dense connections. Zhou et al. argued that using direct skip connections only results in merging feature maps that are not semantically similar [29]. Conversely, using an enriched dense skip connection makes the semantic feature map of the encoder very similar to that of the decoder, which, in turn, makes the optimizer task easier [51]. This also enhances the accuracy of capturing local semantic representation and bridges the semantic gap between the encoder and decoder.

Likewise, in order to enhance feature representation from the encoder, as we did with the decoder, Squeeze-and-Excitation (SE) [52] has been used for the skip connections going out of the encoder, as another form of attention.

Finally, inspired by BASNet [31], the side-output from the decoder was employed for deep supervision during training with UNet++ intermediate outputs. Deep supervision using multiple side-outputs takes advantage of information that is complementary between different scales, intermediates, and prediction maps [53]. In summary, the contributions of our paper are twofold: First, we introduced the TLA module, which benefits from the attention mechanism and spatial normalization on the decoder side. Second, an improved network architecture is proposed with complementary features, including dense connections, Transformer, deep supervision with intermediate side-outputs, and the aforementioned TLA module. These enhancements work hand in hand to achieve a high segmentation accuracy. The proposed model demonstrated superior performance compared to alternative methods, as evidenced by its higher Dice Similarity Coefficient (DSC) and lower Hausdorff Distance (HD) values among all tested datasets from different modalities.

## 2. Materials and Methods

### 2.1. Network Architecture

The objective of the network design is to achieve superior segmentation accuracy by capturing local positioning details, as well as global information. To achieve this, a number of improvements are suggested that complement each other. Most importantly, we introduce an improved Three-Level Attention (TLA) module in the decoder that utilizes the attention map from the Transformer to focus on the most-relevant details, as explained below. The TLA module enhances the two-level attention introduced by TransNorm [42]. Furthermore, we argue that combining the TLA module with UNet++ reduces the semantic gap and leads to improved segmentation accuracy. This is demonstrated by the superior metric scores among multiple datasets, which are discussed later. The overall network architecture is shown in Figure 1. The next subsection provides an overview of the network architecture that includes the dense connections, Transformer, and side-outputs for deep supervision, whereas the latter subsection focuses on the TLA attention module.

#### 2.1.1. Integrating UNet++ to Transformer and Side-Outputs for Deep Supervision

UNet++ and Transformer were merged and utilized in a way similar to TransUNet++ [8]. This was achieved by modifying the network architecture such that the node in the last layer is replaced by two nodes. One node on the encoder side and another on the decoder side and Transformer were added between them, as shown in Figure 1. Note that, for better network clarity, concatenations of the UNet++ dense skip connections on the decoder side were omitted.

Similar to BASNet [31], side-outputs from the decoder are added for deep supervision. These outputs were upsampled and used for deep supervision solely during training, along with other intermediate outputs from UNet++. Empirically, we found that the decoder’s side-outputs enhanced the accuracy. This is depicted in Figure 1 by the dotted red line.

#### 2.1.2. Incorporating Attention in the Network

Inspired by TransNorm [42], a modified three-level gate [42] module comprising an AG [35], convolution, and channel attention [42], called TLA, was developed to focus on important features and suppress irrelevant features, as shown in Figure 2.

The first stage of TLA is the AG. As explained in the Introduction, the AG focuses on crucial features pertinent to a certain task by suppressing feature responses from irrelevant background regions. Oktay et al. [34] argued that this is especially important during downsampling, where small objects may show large shape variability. The AG employed a 1 × 1 channelwise convolution to perform a linear transformation. The two outputs of the feature maps are combined by elementwise addition, followed by ReLU. Another 1 × 1 convolution was employed, followed by sigmoid activation. Moreover, trilinear interpolation was used to scale the output. For TLA, we placed the AG as the first stage immediately after the skip connections to enhance information fusion [49]. Zhang et al. argued that the AG helps remove noise from the background and fixes the blurred boundary that might occur during upsampling [49]. The second attention stage is the channel attention described in [42]. It calculates the average pool and maximum pool for the input and adds them using elementwise addition. Next, it passes this to an MLP layer, followed by sigmoid activation to adaptively recalibrating the weight of each channel. TLA consists of two consecutive layers of convolutions and ReLU and ends with a batch normalization layer as with standard decoder blocks in UNet models. The last stage of TLA is the elementwise multiplication between the coefficients from the Transformer and the feature map. The following is a more-concise mathematical explanation.

First, to suppress irrelevant regions in the image, skip connection features $x\in R^{H^{{}^{\prime }}\times W^{{}^{\prime }}\times C^{{}^{\prime }}}$ and the feature map from the lower encoder blocks and Transformer $z\in R^{H^{{}^{\prime }}\times W^{{}^{\prime }}\times C^{{}^{\prime }}}$ are fed into the AG [35] as shown in Equation (Equation 1).
(1)$$f_{ag}=\left(\sigma \left(\varphi \left(\delta \left(\Theta \left(x\right)\right)+\Theta \left(z\right)\right)\right)\right)⊗x$$
where φ, δ, and Θ denote linear transformations; σ is a sigmoid activation function; and ⊗ represents elementwise multiplication. The outcome is then concatenated with the feature map and used as the input for channel attention. The details of channel attention are explained in [42] and are depicted below in Equation (Equation 2).
(2)$$f_{ca}=\left(\sigma \left(MLP\left(AvgPool\left(F\right)\right)+MLP\left(MaxPool\left(F\right)\right)\right)\right)⊗F$$

This is followed by two convolutions. The output of these two convolutions is represented by $f_{z}$, and the spatial normalization map produced by the Transformer is represented by Equation (Equation 3), where $W_{s}\in R^{1\times H\times W}$.
(3)$$W_{s}=softmax\left(\frac{QK^{T}}{\sqrt{d_{k}}}\right)V$$
are multiplied (elementwise) as shown in (Equation 4).
(4)$$f_{sz}=W_{s}⊗f_{z}$$

Furthermore, SE [52] was added to the skip connection from the encoder to enhance the spatial encoding and feature recalibration, as illustrated in Figure 2. The gray lines represent the attention weights.

**Figure 2 sensors-23-08589-f002:**
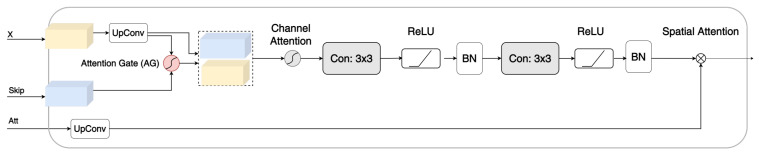
The devised Three-Level Attention module with an Attention Gate (AG), channel attention, and spatial normalization.

### 2.2. Datasets

#### 2.2.1. The Synapse Multi-Organ Segmentation Dataset

The Synapse multi-organ segmentation dataset (https://www.synapse.org/#!Synapse:syn3193805/wiki/217789) accessed on 1 March 2022 from the MICCAI 2015 Multi-Atlas Abdomen Labeling Challenge was utilized in this study. The challenge organizers received medical center and Institutional Review Board (IRB) approval for the release of the data. The dataset contained 3779 axial abdominal Computed Tomography (CT) images. Each CT volume was made up of 85–198 slices, and each slice had a resolution of 512 × 512 px. However, in this implementation, the images were resized to 224 × 224 px for easy comparison with the other models in this paper. The spatial resolution of the voxel was ([0.54-0.54] × [0.98-0.98] × [2.5-5.0]) mm${}^{3}$.

#### 2.2.2. CAMUS Dataset

This study used an ultrasound dataset acquired at the University Hospital of St Etienne (France) and included the regulations set by the local ethical committee of the hospital [25]. The dataset was collected using a GE M5S probe from the GE Vivid E95 ultrasound scanner. The EchoPAC analysis software exported the 2D apical four-chamber and two-chamber view sequences for each patient. The acquisitions were optimized to perform Left Ventricle Ejection Fraction (LV_EF_). The full dataset is available for download (https://camus.creatis.insa-lyon.fr/challenge/) accessed on 30 April 2021. Images from 450 patients were used for training and testing in this study. The dataset contains images of End-Diastole (ED) and End-Systole (ES) for each patient [25].

### 2.3. Evaluation and Metrics

Different evaluation methodologies have been utilized for synapse multi-organ segmentation and CAMUS datasets. To ensure a thorough and fair comparison, the guidelines established by Fu et al. [54] were strictly followed for the assessment of the Synapse multi-organ segmentation dataset. By contrast, when evaluating the CAMUS dataset, the evaluation framework outlined by Leclerc et al. [25] was adopted.

#### 2.3.1. Evaluation of The Synapse Multi-Organ Segmentation Dataset

For the Synapse multi-organ segmentation dataset, the average Dice Similarity coefficient (DSC) and average Hausdorff Distance (HD) were used as statistical validation metrics to evaluate the proposed method on this dataset. The annotation of each image included eight abdominal organs (the aorta, gallbladder, spleen, left kidney, right kidney, liver, pancreas, spleen, and stomach). The dataset comprised 30 cases with 3779 axial abdominal CT images. Following [54,55], a split of 18 (2212 axial slices) and 12 cases was used for training and testing, respectively. Equitable comparisons with alternative approaches were ensured by employing a random selection of training samples for the training dataset and utilizing test images for the test dataset. This random sampling technique was employed to assess the performance disparity between the proposed method and competing models. Similar to [45], we employed a weighted function composed of the Dice loss and cross-entropy loss, which is shown in Equation (Equation 5), whereas the cross-entropy and Dice loss are shown in Equations (Equation 6) and (Equation 7), respectively. Other studies also confirmed that a compound loss function yields a better result than any single loss function [56]. A multiclass version of the Dice loss and cross-entropy variant proposed in [56] was used.
(5)$$\ell_{total}=\lambda_{1}\;\cdot \;\ell_{Dice}+\lambda_{2}\;\cdot \;\ell_{crossentropy}$$
where $\lambda_{1}$ and $\lambda_{2}$ were both set to 0.5 [54]:Cross-entropy loss [56]:(6)$$\ell_{crossentropy}\left(u,y\right)=-\frac{1}{N}\sum_{c=1}^{C}\sum_{i=1}^{N}u_{i}^{c}\log y_{i}^{c}$$
where *C* is the number of classes, *N* denotes the total number of pixels per case, $u_{i}^{c}$ is the ground truth binary indicator of the class label *c* of pixel *i*, and $y_{i}^{c}$ is the corresponding predicted segmentation probability.

Dice loss [56]:


(7)
$$\ell_{Dice}\left(u,y\right)=1-\frac{2\sum_{c=1}^{C}\sum_{i=1}^{N}u_{i}^{c}y_{i}^{c}}{\sum_{c=1}^{C}\sum_{i=1}^{N}u_{i}^{c}+\sum_{c=1}^{C}\sum_{i=1}^{N}y_{i}^{c}}$$


#### 2.3.2. Evaluation of CAMUS Dataset

Similarly, the DSC and 2D Hausdorff distance were utilized as evaluation metrics for the CAMUS dataset. Following [25], the CAMUS dataset was split into ten folds with shuffling to perform standard cross-validation for machine learning techniques. Each fold comprised 45 patients. Each fold was used once for validation, whereas the remaining nine folds formed the training set. Each cross-fold was set to run for 50 epochs, and the results that performed the best on the validation set were saved. Finally, the average of these best-saved results out of the 10-fold is reported. To measure the accuracy of the segmentation output of the Left Ventricle Endocardium (LVEndo), Left Ventricle Epicardium (LVEpi), and Left Atrium (LA) of a given method, the Dice metric and 2D Hausdorff Distance (HD) were used. The evaluation was based on End-Diastole (ED) volume and End-Systole (ES) volume. For each ED and ES volume, the means of LV_Endo_ LV_Epi_ and LA were calculated.

### 2.4. Implementation Details

Training and testing were performed using PyTorch on an NVIDIA RTX A4500 GPU with a 20 GB capacity. The model was trained using a Stochastic Gradient Descent (SGD) optimizer and momentum, with a decaying learning rate. The initial learning rate, momentum, and weight decay were set as 0.005, 0.9, and 0.0001, respectively. Adaptive learning rates were also used in which the learning rate was reduced after a certain number of iterations. The Transformer patch size was set to 16, with a sequence length of 196, and the number of heads was set to 16. To increase the data diversity for the Synapse multi-organ segmentation dataset, augmentation was performed for the training dataset using rotation and image flipping with a probability of 0.5. The number of epochs was set to 150 for the Synapse multi-organ segmentation dataset, similar to [42]. For the CAMUS dataset, no augmentation was performed, following [25]. In order to ensure the fairness of the experiments, we employed an image size of 224 × 224, adhered to identical operating conditions and hyperparameters, and utilized the same training and validation datasets as documented in prior literature.

## 3. Results and Discussion

As mentioned previously, We evaluated the generalization ability of the proposed method with two datasets. The analysis of the two datasets is presented below. We start with the results and discussion of the CT scan Synapse multi-organ segmentation dataset followed by the ultrasound CAMUS dataset. In order to make a fair comparison, we mostly used newer network models for comparison with similar evaluation parameters such as the number of epochs, optimizer, and loss function.

### 3.1. Synapse Multi-Organ Segmentation

Table 1 presents a comparison of the results of the proposed method with those of the other approaches. The table format follows other literature works such as [42,55], where the DSC is the average of the eight abdominal organs (aorta, gallbladder, spleen, left kidney, right kidney, liver, pancreas, spleen, and 201 stomach). The HD is the average of eight abdominal organs. However, unlike the DSC, the HD scores for each organ are not shown.

Our model achieved the best results almost among all organs and consistently outperformed other competing models, with an average DSC and HD of 81.92% and 20.21 mm, respectively. The average DSC increased from 79.13% (Swin-UNet) to 81.92% by 3.53%, while the Hausdorff distance decreased from 21.55 mm (Swin-UNet) to 20.21 mm by 1.34 mm. The result was the average of multiple runs with a standard deviation of 0.57 DSC and 2.90 HD. This improved outcome was due to the incorporation of attention, which was accomplished using our TLA module, as well as the narrowed semantic gap between the encoder and decoder, which was achieved using UNet++ dense connections.

Figure 3 presents the segmentation outcomes of the various approaches on the Synapse multi-organ CT dataset for qualitative comparison. TransUNet and TransNorm were chosen for comparison because their performances were closest to that of our model. Our segmentation result was more precise on the boundary areas and closer to the ground truth. For example, in the first row, TransUNet and TransNorm misclassified the left kidney (red), whereas our model correctly classified it. In the second row, TransUNet misclassified the stomach (white), whereas our model partially classified it. TransNorm misclassified parts of the pancreas (yellow), whereas our model was more similar to the ground truth. In the third row, TransUNet mistakenly classified parts of the liver as the right kidney. In addition, TransNorm did not fully classify the right kidney and overestimated the segment of the stomach. Conversely, our proposed model correctly classified both the liver and stomach. Finally, the image in the fourth row was more challenging for all models, for which both TransUNet and TransNorm over-classified the liver to include parts that were not in the ground truth. In contrast, although our model made little segmentation error in the liver, it was still much better than TransUNet and TransNorm.

This result shows that our model has better segmentation accuracy and is able to delineate the boundaries, even though the segments in the Synapse multi-organ dataset have various shapes and sizes. The model-improved attention using the TLA module helped the network focus on the most-relevant features and avoided misclassifications. TLA was also beneficial for the segmentation task because it enabled better feature fusion between the skip connections and the decoder. Furthermore, deep supervision using multiple side-outputs was able to benefit from feature maps of different scales in a way that they complemented each other. This results in a more-robust prediction when images have different scales [53]. Finally, dense skip connections make the optimization problem between a semantically similar encoder and decoder easier for the optimizer to solve [29]. This, in turn, reduces the possibility of the optimizer getting trapped at a local minimum.

### 3.2. CAMUS Dataset

In order to demonstrate the generalization performance of our model, Table 2 presents comparison results for the CAMUS dataset. Each row in the table presents the results of the network model, followed by its standard deviation. The layout of the table follows a format similar to that in [25]. As can be observed, the proposed method outperformed the other approaches in the DSC in both endocardium LV_Endo_ and epicardium LV_Epi_ respectively. Furthermore, our model produced the best HD for both the endocardium LV_Endo_nd epicardium LV_Epi_ indicating that the results of our model are more generalizable across diverse datasets. As indicated in Table 2, our model achieved a DSC score of 92.52 and an HD score of 11.04 on the End-Diastole (ED) images, while on the End-Systole (ES) images, our model achieved a score of 92.64 on the DSC and a score of 12.35 on the HD. This shows that our model consistently achieves superior outcomes using different matrices and datasets.

It might be surprising that Swin-UNet performed poorly compared to TransUNet on the CAMUS dataset, although Swin-UNet usually outperforms models with full convolutions or a combination of Transformer and convolutions [55]. This outcome can be attributed to the following reasons. First, the CAMUS dataset is smaller than the Synapse dataset. Because Swin-UNet is purely a Transformer-based model, it requires more training samples than TransUNet, which is a combination of Transformer and convolution. Second, for a fair comparison, both models were trained for 50 epochs. As TransUNet employs a CNN, it has a faster inference speed owing to its inductive bias [59]. It should be noted that a model based solely on Transformer could benefit more from a larger number of iterations.

In addition, for visual comparison, the result is presented in Figure 4 in order to highlight the significance of the attention mechanism. Note that, in the first row, Swin-UNet produced inferior results compared with TransUNet and our model. Swin-UNet segmentation resulted in a zigzag contour with poor quality. Although both TransUNet and our model produced good results, our model produced a better segment for $LV_{Endo}$ (the $LV_{Endo}$ produced by TransUNet was smaller than the ground truth). The ultrasound image in the second row was challenging for the segmentation of all models, including ours. This is because of the low contrast and blurred boundaries between the different segments. It is clear that our model performed better than Swin-UNet did. In addition, comparing our model to TransUNet, note how $LV_{Epi}$ in our model was similar to the ground truth, while $LV_{Epi}$ in TransUNet was smaller than the ground truth (the line touches the edge in both our model and the ground truth, while being a little bit far in TransUNet). In the third row, TransUNet segmentation mistakenly showed parts of $LV_{Endo}$ inside the LA walls. Moreover, Swin-UNet produced low-quality zigzagged contours for $LV_{Epi}$, $LV_{Endo}$, and LA. The segmentation of our model was similar to the ground truth. Finally, in the last row, both our model and TransUNet performed well, except Swin-UNet, which again produced zigzagged segment boundary lines. Moreover, our model performed slightly better than TransUNet in terms of $LV_{Epi}$, which is more similar to the ground truth in our model.

## 4. Ablation Study

We conducted ablation experiments to demonstrate the justification of each contribution and justified the rationale of the design choices for the overall performance. The results, recorded on five metrics, are presented in Table 3 for the Synapse multi-organ CT dataset. The comparison is presented in Table 3 for the Synapse multi-organ CT dataset to avoid repetition. In contrast, the CAMUS dataset was used to test the behavior of different optimizers, as shown in Figure 5 and Figure 6.

Our results revealed the significance of the proposed improvements compared with the baseline. The baseline was considered to be TransNorm with a UNet-like architecture, Transformer, and spatial normalization mechanism. First, as shown in Table 3, adding UNet++ improved both the DSC and HD metrics. As mentioned previously, UNet++’s dense connections aid in reducing semantic gaps. However, we noticed that integrating UNet++ achieved excellent results when combined with deep supervision using side-outputs. The side-output connections were helpful in enriching the deep supervision during the training phase. The Three-Level Attention contributed to an improvement in the DSC score. Finally, adding a Squeeze-and-Excitation (SE) block to the encoder was helpful for both the DSC and HD scores.

Furthermore, in order to understand the effect of experimenting with models of different parameter sizes, an ablation study was performed on the model scale, following the literature [45], result shown in Table 4. The base model had the following hyperparameters: the hidden size was set to 12; the number of layers was set to 768; the MLP size was set to 3072; the number of heads was set to 12. On the other hand, the hidden size of the large model was set to 24; the number of layers was set to 1024; the MLP size was set to 4096; the number of heads was set to 16. The results indicated that large models perform better, as expected, although they require more computation [60].

To investigate the effects of using different optimizers, we experimented with different optimizers using the CAMUS dataset. Figure 5 and Figure 6 show the plots for the four optimizers: SGD, Adam, AdamW [61], and AdaDelta. For clarity, the training and validation losses are shown in different figures. Although there are many interesting and newer algorithms, such as Sharpness-Aware Minimization (SAM) [62], they are among the most-commonly used [63]. SGD utilizes one sample randomly in each iteration to update the gradient instead of exactly calculating the value of the gradient [63]. The Adam optimizer is one of the most-widely used stochastic optimization algorithms. It is similar to SGD, but introduces an adaptive learning rate for each parameter. It dynamically recalculates the learning rate of each parameter using first-order and second-order moment estimations of the gradient [1]. It has been noted that models trained with Adam have been observed to not generalize well as SGD [61]. Hence, the AdamW stochastic optimization method was introduced, which modifies the typical implementation of the weight decay in Adam by decoupling the weight decay from the gradient update. AdaDelta is an improvement over other optimizers, such as AdaGrad and RMSProp [63], to solve the vanishing gradient problem. The idea is to focus on the gradients in a window by using an exponential moving average [62].

Figure 5 shows the behavior of each optimizer during our training experiment. Both SGD in Figure 5a and AdaDelta in Figure 5d have a smooth learning curve, whereas Adam in Figure 5b and AdamW in Figure 5c exhibit an oscillating behavior.

Similarly, Figure 6 displays plots of the Dice loss for the validation dataset. Again, Adam in Figure 6b and AdamW in Figure 6c exhibit a fluctuating behavior, where AdamW exhibits extreme oscillation, while SGD in Figure 6a and AdaDelta in Figure 6d are the smoothest.

## 5. Conclusions

This paper presented an improved network architecture for medical image segmentation. Our proposed method incorporates a combination of a CNN and Transformer to extract global semantic information and local features. Furthermore, we proposed a TLA module with dense UNet++ connections and supervision using side-outputs to enhance the localization ability and boundary delineation, leading to more-accurate segmentation results. Through rigorous evaluation of the CT and ultrasound datasets, our proposed model showed superior performance compared to existing methods. We hope this work contributes to advances in medical image segmentation and provides a more-reliable tool for clinical applications. Future research could build on our proposed improvements to further enhance the accuracy and robustness of medical-image-segmentation models.

## Figures and Tables

**Figure 1 sensors-23-08589-f001:**
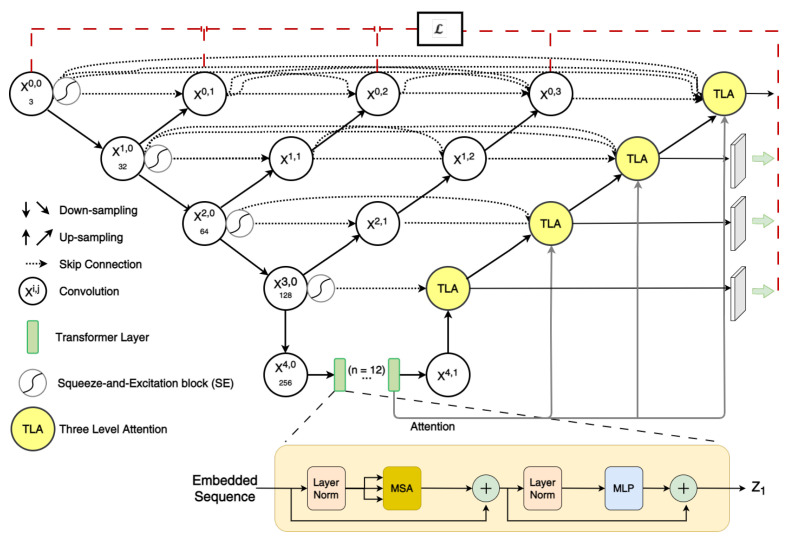
The proposed network architecture includes a Transformer, deep supervision with side-output, a Three-Level Attention module (TLA), and SE. Note that the concatenation in the main network diagram has been omitted for clarity. In addition, the 1×1 convolution for altering the number of channels is omitted for clarity. Details of the TLA module are shown in Figure 2.

**Figure 3 sensors-23-08589-f003:**
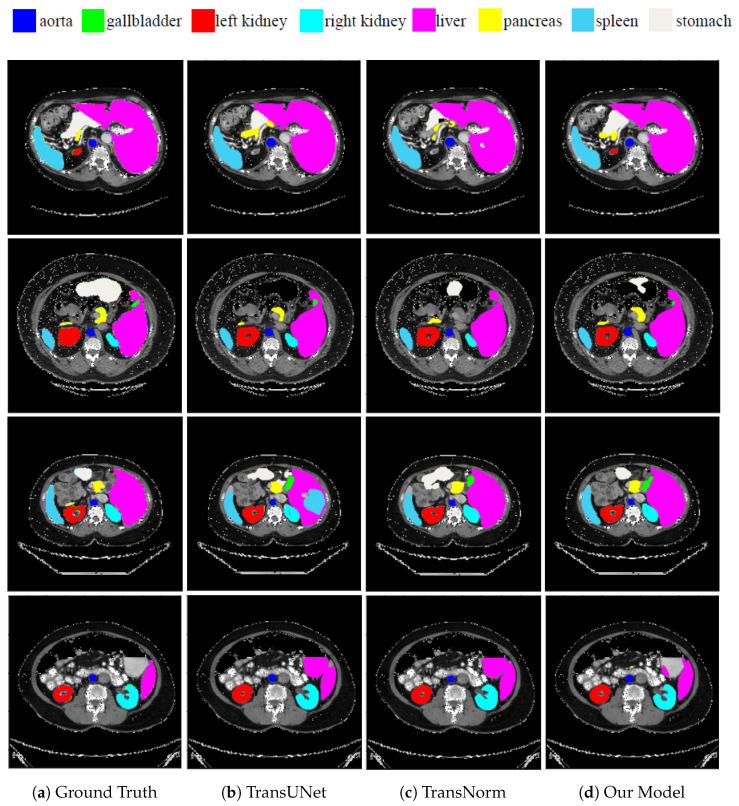
Qualitative comparison of random CT slices from the Synapse dataset. From left to right: (**a**) ground truth, (**b**) TransUNet predication, (**c**) TransNorm predication, and (**d**) our model’s predication.

**Figure 4 sensors-23-08589-f004:**
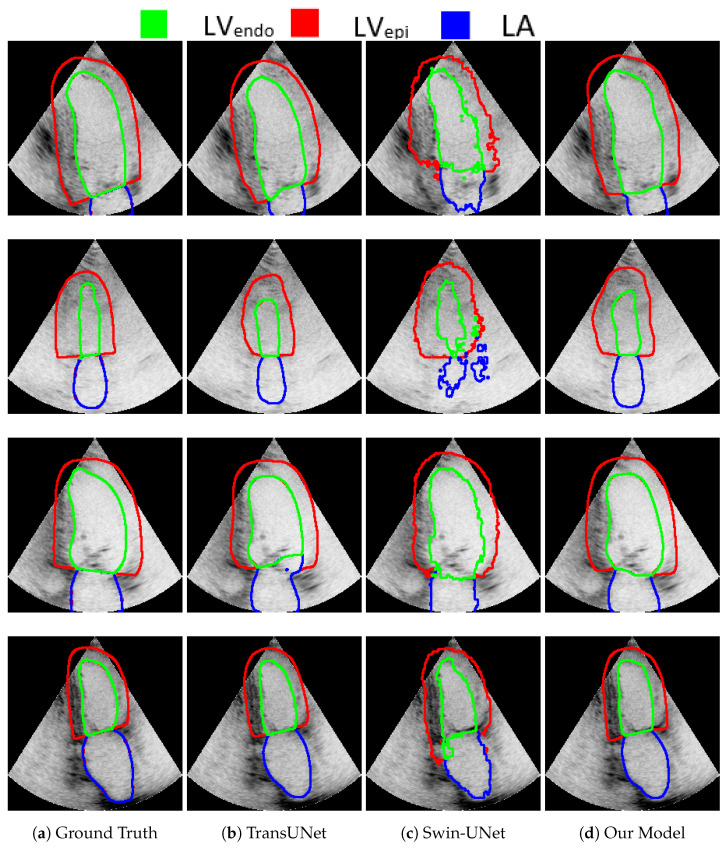
Qualitative comparison of random images from the CAMUS dataset. Endocardium and epicardium of the left ventricle and left atrium wall are are indicated by blue, green, and red, respectively. From left to right: (**a**) ground truth, (**b**) TransUNet predication, (**c**) TransNorm predication, and (**d**) our model’s predication.

**Figure 5 sensors-23-08589-f005:**
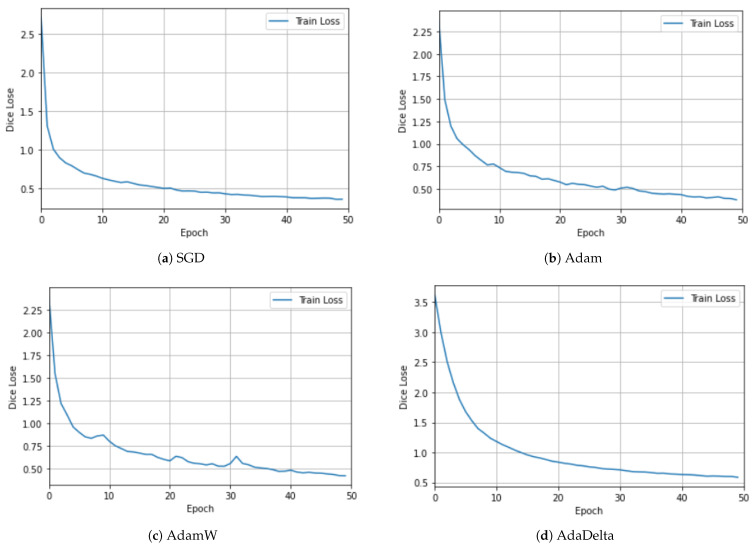
Ablation study using different optimizers with the CAMUS dataset for the training dataset. Both SGD and AdaDelta had a smooth learning curve, whereas Adam and AdamW exhibited an oscillating behavior. The best results were achieved with SGD, Adam, and AdamW, with similar magnitudes.

**Figure 6 sensors-23-08589-f006:**
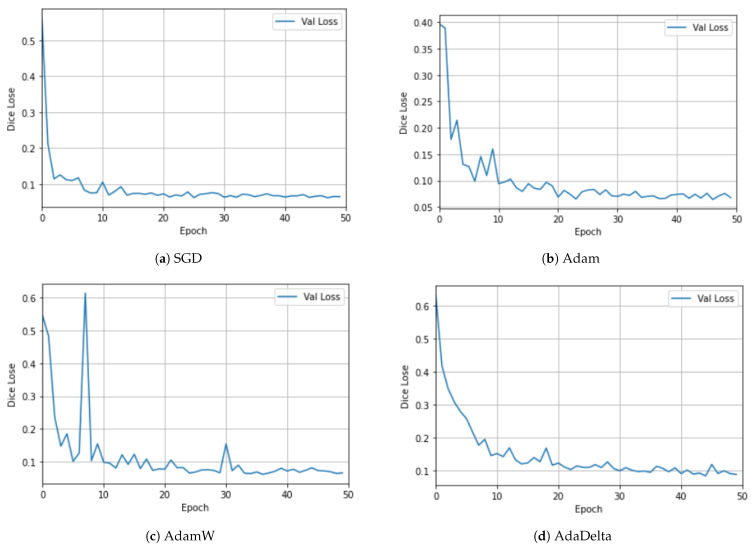
Ablation study using different optimizers with the CAMUS dataset for the validation dataset.

**Table 1 sensors-23-08589-t001:** Comparison of the segmentation accuracy on the Synapse multi-organ CT dataset (average Dice score % and average Hausdorff distance in mm % and Dice score % for each organ). The best scores are shown in bold.

Model	DSC↑	HD↓	Aorta	Gallb.	Kidney (L)	Kidney (R)	Liver	Pancreas	Spleen	Stomach
R50 UNet [45]	74.68	36.87	87.74	63.66	80.6	78.19	93.74	56.9	85.87	74.16
UNet	76.85	39.70	**89.07**	69.72	77.77	68.6	93.43	53.98	86.67	75.5
R50 ViT [45]	71.29	32.87	73.73	55.13	75.8	72.2	91.51	45.99	81.99	73.95
TransUNet [45,55]	77.48	31.69	87.23	63.13	81.87	77.02	94.08	55.86	85.08	75.62
UCTransNet [57]	78.99	30.29	−	−	−	−	−	−	−	−
Swin-UNet [55]	79.13	21.55	85.47	66.53	83.28	79.61	94.29	56.58	**90.66**	76.60
TransNorm [42]	78.40	30.25	86.23	65.10	82.18	78.63	94.22	55.34	89.50	76.01
MT-UNet [58]	78.59	26.59	87.92	64.99	81.47	77.29	93.06	59.46	87.75	76.81
Proposed Method	**81.92**	**20.21**	89.01	**70.39**	**86.04**	**82.83**	**95.09**	**62.32**	90.02	**78.33**

**Table 2 sensors-23-08589-t002:** Segmentation accuracy of the evaluated methods on the ten test folds.

Model	ED		ES	
	**DSC↑**	**HD↓**	**DSC↑**	**HD↓**
UNet	91.40	11.89	91.42	13.27
(SD)	±0.9562	±1.0046	±0.5941	±0.7397
UNet++	92.04	11.53	92.32	12.65
(SD)	±0.7336	±0.5889	±0.6699	±0.6788
TransUNet	91.23	12.06	91.42	13.37
(SD)	±0.4414	±0.6234	±0.6660	±0.5664
Swin-UNet	84.34	16.33	85.71	16.86
(SD)	±0.8296	±0.7783	±1.1337	±0.8415
TransNorm	90.18	11.15	90.87	13.47
(SD)	±0.8957	±1.7352	±0.4438	±0.6467
**Proposed Method**	**92.52**	**11.04**	**92.64**	**12.35**
(SD)	±0.5068	±0.5302	±0.7081	±0.5990

ED stands for End-Diastole, and ES stands for End-Systole. SD stand for Standard Deviation. The score is the average segmentation score of Ventricle Endocardium (LV_Endo_) Epicardium (LV_Epi_) and the Left Atrium (LA).

**Table 3 sensors-23-08589-t003:** Contribution of each change to the overall performance of the Synapse multi-organ CT dataset.

Methods	DSC↑	HD↓
Baseline	78.40	30.25
Baseline + “UNet++”	79.37	30.46
Baseline + “UNet++” + side output (deep supervision)	81.49	21.59
Baseline + “UNet++” + side output + TLA module	81.80	26.07
Baseline + “UNet++” + side output + TLA module + SE	**81.92**	**20.21**

**Table 4 sensors-23-08589-t004:** Ablation study on the model scale.

Model Scale	DSC↑	HD↓	Aorta	Gallb.	Kidney (L)	Kidney (R)	Liver	Pancreas	Spleen	Stomach
Base	81.92	20.21	89.01	70.39	86.04	**82.83**	**95.09**	62.32	90.02	**78.33**
Large	**82.69**	**16.41**	**89.80**	**73.67**	**86.09**	82.08	95.01	**65.68**	**92.81**	76.41

## Data Availability

The Synapse multi-organ segmentation dataset is available at https://www.synapse.org/#!Synapse:syn3193805/wiki/217789 accessed on 1 March 2022, and the CAMUS dataset is available at https://camus.creatis.insalyon.fr/chalenge/ accessed on 30 April 2021.

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
