# Peer review of "Improved UNet with Attention for Medical Image Segmentation"

_sensors, 2023, doi:10.3390/s23208589_

Round 1

Reviewer 1 Report

This article proposes a new model to address the challenges of medical image segmentation caused by blurry boundaries, low contrast in organizational environments, small size, multiple types, and speckle noise and attenuation under ultrasound conditions. This model combines the advantages of CNN and Transformer, and improves the network architecture, mainly by introducing a new attention module TLA and redesigning the skip connection between the encoder and decoder. Several review comments are as below:

1. Lack of comparison between methods and more advanced methods;

2. The writing in English needs to be strengthened;

3. The experimental part needs to be described in more detail;

4. The improvement of the model in Figure 2 needs to be described comprehensively

The writing in English needs to be strengthened

Author Response

Dear Reviewer;

We made a combined document for all reviewer's responses. Please, see the attached PDF file. Please, go to reviewer 1.

Thank you!

Reviewer 2 Report

The article is devoted to solving the problem of medical image segmentation. The topic of the article is relevant. The structure of the article does not correspond to that accepted in MDPI for research articles (Introduction (including analysis of analogues), Models and methods, Results, Discussion, Conclusions). The level of English is acceptable. The article is easy to read. The figures in the article are of acceptable quality. The article cited 64 relevant sources. The References section is designed carelessly.

The following comments and recommendations can be formulated regarding the material of the article:

1. Currently, U-net has a dominant position in solving image segmentation problems, especially in the field of medical imaging. Among most of the U-nets proposed so far, convolutional neural networks (CNNs) are widely used as basic structures. However, CNNs can only effectively use short-range (“last 100 meters”) (or local) information due to the small size of the convolution kernel, which prevents them from sufficiently exploring data in problems containing components with long-range dependencies. Transformers can effectively explore information over long distances, but are not reliable enough to work with data close up, as CNNs do. To exploit the capabilities of CNNs to compensate for the shortcomings of transformers and on the other hand in image segmentation tasks, in the 2020s, Chen et al proposed TransUNet, which is also the first image segmentation model built on a transformer. Since then, tons of TransUNet tuning work has been published. I ask the authors to explain why another work in this direction is needed?

2. The authors mention UNet++. It is a powerful architecture for medical image segmentation. This architecture is essentially a deep learning encoder-decoder network in which the encoder and decoder subnetworks are connected by a series of nested layers. The redesigned layers aim to reduce the semantic gap between the feature maps of the encoder and decoder subnets. Why was this model not enough?

3. Medical image segmentation faces many problems, which cause the quality of the segmentation process to be unsatisfactory. In most cases, the image under study contains noise, distortion, and texture areas similar to the areas belonging to the object under study. All this complicates the process of selecting objects and correctly displaying their boundaries, so contouring and segmentation algorithms play a very important role in the automated processing process. How did the authors take this into account?

Author Response

Dear Reviewer;

We made a combined document for all reviewer's responses. Please, see the attached PDF file. Please, go to reviewer 2.

Thank you!

Reviewer 3 Report

The paper introduces a new model that combines the strengths of both CNNs and Transformers for automatic medical image segmentation. The proposed U-Net architecture is evaluated on two datasets from different modalities: a CT scan dataset and an ultrasound dataset. The experimental results consistently demonstrate improved prediction performance compared to the traditional U-Net across different datasets. Overall, the paper presents a novel hybrid architecture combining CNN and Transformer models. The proposed Three-Level Attention module, redesigned skip connections, and deep supervision contribute to improved performance. The evaluation on diverse datasets strengthens the validity of the findings.

(1)The references cited in Table 1 are both derived from preprint papers, which have not undergone peer review and therefore possess a lower level of reliability. It is advisable to substitute these with published articles from reputable scholarly journals.

(2)Table 2 compares multiple models to highlight the differences, and the results indicate that the proposed new method outperforms others. However, it is worth noting that in many cases, the scores exhibit only minimal differences. In such instances, these marginal improvements may lack practical significance. It is recommended that the authors provide statistical comparison results to determine whether the differences between different methods are statistically significant. This would help to provide a more comprehensive evaluation of the performance disparities among the compared models.

(3)The results depicted in Figure 5 and 6 suggest that the model exhibits a certain degree of overfitting. After approximately 20 iterations, the performance on the validation set reaches a plateau, indicating that further training may result in oscillations rather than significant improvement. This observation may explain why the authors mentioned a total of 150 iterations in the implementation details but only plotted the results for 50 iterations.

To address this issue, it is recommended to undertake further optimization of the model parameters to mitigate overfitting. Techniques such as regularization methods or early stopping strategies could be employed to prevent the model from excessively fitting the training data. Conducting experiments with different hyperparameter settings and evaluating their impact on the model's generalization ability would also contribute to improving overall performance and reducing overfitting tendencies.

Author Response

Dear Reviewer;

We made a combined document for all reviewer's responses. Please, see the attached PDF file. Please, go to reviewer 3.

Thank you!

Round 2

Reviewer 2 Report

I formulated the following comments to the previous version of the article:

1. Currently, U-net has a dominant position in solving image segmentation problems, especially in the field of medical imaging. Among most of the U-nets proposed so far, convolutional neural networks (CNNs) are widely used as basic structures. However, CNNs can only effectively use short-range (“last 100 meters”) (or local) information due to the small size of the convolution kernel, which prevents them from sufficiently exploring data in problems containing components with long-range dependencies. Transformers can effectively explore information over long distances, but are not reliable enough to work with data close up, as CNNs do. To exploit the capabilities of CNNs to compensate for the shortcomings of transformers and on the other hand in image segmentation tasks, in the 2020s, Chen et al proposed TransUNet, which is also the first image segmentation model built on a transformer. Since then, tons of TransUNet tuning work has been published. I ask the authors to explain why another work in this direction is needed?

2. The authors mention UNet++. It is a powerful architecture for medical image segmentation. This architecture is essentially a deep learning encoder-decoder network in which the encoder and decoder subnetworks are connected by a series of nested layers. The redesigned layers aim to reduce the semantic gap between the feature maps of the encoder and decoder subnets. Why was this model not enough?

3. Medical image segmentation faces many problems, which cause the quality of the segmentation process to be unsatisfactory. In most cases, the image under study contains noise, distortion, and texture areas similar to the areas belonging to the object under study. All this complicates the process of selecting objects and correctly displaying their boundaries, so contouring and segmentation algorithms play a very important role in the automated processing process. How did the authors take this into account?

The authors responded to all my comments. I found their answers quite convincing. I support the publication of the current version of the article. I wish the authors creative success.